# Sensitivity Improvement to Active Piezoresistive AFM Probes Using Focused Ion Beam Processing

**DOI:** 10.3390/s19204429

**Published:** 2019-10-12

**Authors:** Piotr Kunicki, Tihomir Angelov, Tzvetan Ivanov, Teodor Gotszalk, Ivo Rangelow

**Affiliations:** 1Faculty of Microsystems, Electronics and Photonics, Wroclaw University of Science and Technology, 50-372 Wroclaw, Poland; teodor.gotszalk@pwr.edu.pl; 2Department of Micro- and Nanoelectronic Systems (MNES), Institute of Micro and Nanoelectronics, Ilmenau University of Technology, Gustav-Kirchhoff-Str.1, 98693 Ilmenau, Germany; tihomir.angelov@tu-ilmenau.de (T.A.); Tzvetan.ivanov@tu-ilmenau.de (T.I.)

**Keywords:** MEMS, silicon cantilever, FIB modifications, FEM

## Abstract

This paper presents a comprehensive modeling and experimental verification of active piezoresistive atomic force microscopy (AFM) cantilevers, which are the technology enabling high-resolution and high-speed surface measurements. The mechanical structure of the cantilevers integrating Wheatstone piezoresistive was modified with the use of focused ion beam (FIB) technology in order to increase the deflection sensitivity with minimal influence on structure stiffness and its resonance frequency. The FIB procedure was conducted based on the finite element modeling (FEM) methods. In order to monitor the increase in deflection sensitivity, the active piezoresistive cantilever was deflected using an actuator integrated within, which ensures reliable and precise assessment of the sensor properties. The proposed procedure led to a 2.5 increase in the deflection sensitivity, which was compared with the results of the calibration routine and analytical calculations.

## 1. Introduction

Piezoresistive cantilevers have been used more and more frequently as versatile sensors in atomic force microscopy (AFM) [1]. In 1994, I.W. Rangelow et al. [2] developed a cantilever with a complete Wheatstone bridge integrated with the spring beam. The proposed solution allowed one to detect tip deflection with the resolution of 0.2 Å, which corresponded with the force detection resolution of 420 pN in the bandwidth of 1 kHz [3]. The probe fabrication technology described here is based on standard silicon wafers, which makes it possible to manufacture cost effective AFM probes fabrication with high yield for almost all scanning probe (SPM) technologies [4]. In this way, probes with stiffness in the range of 0.1–40 N/m and with resonant frequencies in the range of 40–800 kHz have been fabricated. The probes typically show a deflection sensitivity of (δR/R)/z = 7.8 × 10^8^ 1/nm and a force sensitivity of (δR/R)/F = 0.8 × 10^−6^ 1/nN.

The temperature drift compensation and higher output of the piezoresistive detector have been achieved by a highly symmetrical on-chip Wheatstone bridge configuration [5]. Furthermore, the relatively high repeatability achievable by today’s CMOS fabrication processes allows for resistor manufacturing of nearly equal resistances, which greatly improves measurement accuracy [6]. The next breakthrough in the technology of the piezoresistive SPM cantilevers was proposed in 2004 when a system integrating not only a piezoresistive deflection detector but also a deflection actuator was developed [7]. The proposed methodology of the so called active SPM cantilevers is very attractive as it eliminates the need for application of a complex piezoactuator setup, which is commonly used in passive cantilever SPM technology. The active SPM piezoresistive cantilevers proved to be sensitive enough for study of many surface properties down to the atomic scale [8].

The lack of external laser beam focusing and alignment setup and optical deflection sensing elements simplifies significantly the design of atomic force microscope for large samples imaging and operation in liquids or ultra-high vacuum (UHV) environments. The integration of the deflection actuator makes it also possible to fabricate arrays of cantilevers, which results in the throughput increase, as surface imaging can be done using many cantilevers operating in parallel. Because the deflection actuator actuates the motion of only the micromechanical cantilever whose mass is very tiny, the response time is limited by the beam resonance frequency (which is often in the range of 100 kHz) and not by the resonance frequency of the bulky microscope piezoactuator (which is in the range of several kHz). However, additional modifications of the piezoresistive active cantilevers architecture must be introduced, as many non-standard AFM imaging investigations, including measurements of tall structures, are foreseen in the near future. Thereby, in this paper we describe a method of piezoresistive active cantilever measurement sensitivity improvement which is based on focused ion beam (FIB) technology. The novelty of the proposed methodology relies on the application of the defined excitation generated by the integrated deflection actuator to monitor the changes of the piezoresistive detector output. Owing to the proposed modification of the beam mechanical structure we are able to increase the tip deflection sensitivity 2.5 times by reducing the sensors resonance frequency only by 4.5%. The calibrated noise level is 0.037 nm/√Hz, which makes it possible to perform high resolution and high speed surface imaging in air or vacuum. The developed FIB technology of the piezoresistive active cantilever tuning is associated with finite element modeling (FEM) and development of high resolution metrology methodology. Finally, we calibrated, which means we described the piezoresistive sensor deflection sensitivity in a quantitative way.

## 2. Theoretical Background

The piezoresistive behavior of silicon was first documented by Smith in 1954 [9]. Single crystal silicon has been widely used as a stress sensor because the changes of the piezoresistor resistance related to the changes of the material resistivity are much bigger than the changes to the resistance related to the changes of the resistor geometry. Furthermore, because the piezoresistive effect depends on the crystal directions, there are many possibilities to arrange the piezoresistive strain sensors setup in a way ensuring the highest response of the deflection sensitivity. In the piezoresistive cantilevers the mechanical stress sensor is integrated with the silicon spring beam. In this setup, when the cantilever is bent by the force *F*, the mechanical stress *σ*_1_ is observed on the beam surface and it can be calculated according to the formula:(1)$$\sigma_{1}=\frac{d\left(l-x\right)}{2I_{z}}F,$$
where *I_z_* is the beam inertia moment, *l* the cantilever length, *x* is the position along the beam (*x* = 0 at the beam supporting point) and *d* is the cantilever thickness. The relative change of the resistance of the longitudinally and transversally stressed piezoresistors forming the Wheatstone bridge can be calculated in accordance with
(2)$$\frac{\Delta R}{R}\approx \frac{\Delta R_{T,i}}{R_{T}}\approx -\frac{\Delta R_{L,i}}{R_{L}}$$
and
(3)$$\frac{\Delta R}{R}\approx \frac{\Delta R_{T,i}}{R_{T}}\approx -\frac{\Delta R_{L,i}}{R_{L}}$$
where *π_L_* and *π_T_* are the longitudinal and the transversal piezoresistive coefficients of silicon. For the Wheatstone bridge configuration it can be assumed that
(4)$$\frac{\Delta R}{R}\approx \frac{\Delta R_{T,i}}{R_{T}}\approx -\frac{\Delta R_{L,i}}{R_{L}}$$
and as a result, the output of the deflection sensor being formed by four piezoresistors is four times bigger than the single piezoresistor setup. The analysis of the above formulas shows clearly that the stress decreases linearly along the cantilever beam. Therefore, modification of the mechanical beam using focused ion beam (FIB) technology, so that the entire piezoresistor structure will receive uniform and concentrated mechanical stress, will result in the higher response of the deflection sensor. In our experiments we observed the cantilever response when the structure was deflected using the integrated thermomechanical deflection actuator [10]. When the microheater is electrically biased, Joule heat is dissipated in the spring beam according to the formula:(5)$$\begin{array}{l} P_{H}=\frac{1}{R_{H}}{\left[U_{dc}+U_{ac}\sin\omega_{H}t\right]}^{2}=\\ =\frac{1}{R_{H}}\left[\begin{array}{l} \left(U_{dc}^{2}+\frac{1}{2}U_{ac}^{2}\right)+\\ +2U_{ac}U_{dc}\sin\omega_{H}t-\left(\frac{1}{2}U_{ac}^{2}\cos(2\omega_{H}t)\right) \end{array}\right]=\\ =\left[P_{dc}+P(\omega_{H})+P(2\omega_{H})\right] \end{array}$$
where *R_H_* is the resistance of the thermal deflection actuator, *ω_H_* is the frequency of the voltage applied to the thermal deflection actuator, *U_dc_* is the bias voltage, *U_ac_* is the alternating voltage, *P_dc_* is the power caused by bias voltage and *t* is time. Due to the differences between coefficients of thermal linear expansion of the materials forming the cantilever beam, the tip is forced to deflect when heat is transported through the beam. According to Equation (5), static and resonance cantilever deflection can be excited by varying the heating current. In all our experiments we biased the microheater in the same way, which stabilized tdhe test excitation conditions during the FIB based optimization of the mechanical structure.

In our experiments we calibrated the sensitivity of the piezoresistive cantilever analyzing its thermomechanical vibration noise. In general, when the cantilever dimensions are known, the determination of the cantilever stiffness based on the resonance frequency measurement method is a convenient technology [11]. However, if the micromechanical sensor structure is more complex the more convenient technology is to rely on the analysis of thermomechanical cantilever vibration [11]. The piezoresistive cantilever is a mechanical system which can be modelled as a simple harmonic oscillator with a response function influenced by vibrations at the resonance frequency. The vibration at this frequency forms a mechanical degree of freedom driven by 1⁄2 k_b_T of energy if placed in a thermal bath. The corresponding response to the thermal excitation, more exactly the resulting noise power spectral density of the cantilever displacement 〈z_th_〉^2^, is the superposition of contributions from all modes and can be calculated on the basis of the Nyquist theory [11]. If the simple harmonic oscillator model is valid, the internal damping of the cantilever is small, 〈z_th_〉^2^ is given by: 〈z_th_〉^2^ = (2k_B_ T)/k, where k is the cantilever stiffness, T is the temperature and k_B_ is the Boltzmann constant. When thermomechanical vibrations are measured using a high resolution interferometer it is quite straightforward to calculate the cantilever stiffness. In the case of the piezoresistive sensors the described methodology has an additional advantage. When the electrical output of the piezoresistive detector is recorded in parallel with the measurement of the mechanical structure vibration, it is possible to calibrate the deflection sensor sensitivity as well [12]. It should be noted that, in contrast to other technologies based on sensor loading, the described technology offers the highest throughput and accuracy.

## 3. Modelling and Simulation

The implications of the varying composite structure of the beam were analyzed with a finite element (FE) model. Four different geometrical models were chosen to optimize displacement outputs of the beam. Geometric and material properties used for the simulations are shown in Table 1 and Table 2. For all of the simulations the force acting on the tip of the cantilever was kept constant 30 uN.

For the numeric analysis of the stress distribution in the cantilever beam and its corresponding displacement due to the force acting on the tip of the cantilever, the commercial finite element analysis (FEA) software COMSOL Multiphysics 4.4 was used. Considering a steady-state condition, each solution was based on a mesh with about 11,000 elements. For the simulations fixed boundary was the base of the cantilever. Between the Si and the Al layers there was a SiO_2_-Si_3_N_4_ layer. Figure 1 shows the stress concentrations obtained in the area of the piezoresistive readout for the four different geometries. During the simulations obtained deflection of the free end of the cantilever was up to 0.9 um. The COMSOL package used a system of differential equations and after the discretization in space the partial differential equations were obtained. As a result, the implementation of a model reduction algorithm usually depends on a particular sparse solver. We predicted the behavior of piezoresistive cantilevers by coupling thermomechanical actuation with the use of the FEM program COMSOL. Dynamic and static deflection measurements done in SEM on a realized cantilever confirmed the results of the simulation. In addition, the static deflection behavior of the active cantilever was investigated.

Figure 2 shows a cross-section of the mechanical stress calculated in the center of the piezoresistive cantilever for the structure before and after three steps of the FIB modification. The area in which the piezoresistors were integrated is marked as well. It can be seen that, after step 3 of the structure modification, the mechanical stress increased in the area of the piezoresistors, which indicates clearly that the bending of the cantilever occurred mostly at those locations. 

## 4. Fabrication

We fabricated piezoresistive AFM probes with integrated Si tips, which are formed at the beginning of the cantilever micromachining process. The fabrication procedure for the cantilever beam is similar to that previously presented [13]. Silicon on insulator wafer (SOI) was used for plasma etching of the tips. The thickness of the top layer may be varied depending on the Si tip required height. If the silicon tip has to be integrated on the cantilever, thermal oxide needs to be grown and patterned to form a mask which is subsequently used for wet etching of the Si tip. After a standard RCA clean, an 8000 Å thick oxide was grown. This film was patterned and the resist mask over the oxide was used as a mask for the boron doping. Due to a repeat use of a resist mask the piezoresistors were configured in the oxide layer and boron doping followed by growth of a passivating oxide layer. Aluminum/Magnesium for the contacts to the piezoresistors and the metal layer, forming the thermomechanical actuators, was then deposited. To form the cantilever beam and to cut the single sensor chip employing a reactive ion etching step, a thick resist mask was used. The cantilevers presented in this paper exhibited thickness in the range of 2.5 µm to 10 µm and the average depth of the resistor was 0.100 nm. The piezoresistors were p type with a sheet resistance of 200 a. The cantilever was oriented along the (110) crystallographic axis of the silicon, where the piezoresistive coefficients of silicon crystal are maximal.

The piezoresistors in the Wheatstone bridge configuration were formed using Boron evaporation and rapid thermal annealing (RTA) at 1100 °C for 15 s, where the thermal effects are minimized subject to sustainment of the sensitivity. The piezoresistor doping becomes a shallow and uniform step profile of depth. Better performance of piezoresistors is obtained when the piezoresistive layer is much shallower relative to the total cantilever thickness [14].

## 5. FIB Modifications

The rectangular-shaped silicon active piezoresistive SPM cantilever was modified using a SEM/FIB FEI NanoLab Helios 600i dual beam system with an electrical multipin flange for connecting of the dedicated measurement and control electronics. The response of the piezoresistive deflection sensor was monitored during the entire procedure, which involved a sequence of three FIB milling steps. The FIB milling was done with Gallium ion column on the back side of the cantilever in order to avoid the ion implanting in the piezoresistor and microheater cantilever areas. The investigations of the cantilever resonance behavior were conducted at a pressure of 10^−6^ mbar. For the thermal probe actuation and read-out of the deflection signal dedicated electronics were designed and integrated in the microscope chamber to enable the in-situ characterization (Figure 3a). The piezoresistive bridge was supplied with symmetrical, low-noise and stable bias voltage of ±1 V. The signal from the deflection sensor was amplified with the gain of 13,000 V/V, the thermal input voltage noise of the input amplifier was 3 nV/√Hz and the bandwidth was 2 MHz. In order to record signals from the measurement electronics placed inside the FIB/SEM microscope chamber, a lock-in amplifier Stanford SR510 and a multimeter Keithley 2000 were used (Figure 3b). The resonance curves were recorded by the microheater current sweeping. As the signal source ensured stable and precise microheater biasing, a single channel Tectronix AFG3101C function generator was applied. After the FIB modification of the cantilever structure, the sensor thermomechanical noise vibration was recorded using a SP-100 SIOS interferometer. The recorded data were analyzed using software designed in the LabView environment and analogue to digital and digital to analogue conversion cards in order to determine the beam stiffness and the deflection sensitivity.

The FIB milling procedure was performed with 30 kV accelerating voltage with current of 9.3 nA in three steps. The milling patterns defined in simulations were applied. Figure 4 shows the results of every step of milling. After every modification step resonance frequency and the relative piezoresistive cantilever sensitivity, induced in the defined and maintained excitation conditions were recorded (Table 3). The information obtained in this way enables one to calculate the shift of the resonance frequency which was a measure for the stiffness change. In Table 3 we also summarized the change of the stress which will be received by the Wheatstone bridge piezoresistors calculated based on the FEM, whose results are shown in Figure 4. Assuming that the structure of the beam is rectangular, the change in the cantilever stiffness can be estimated. The modifications performed lead to the cantilever softening of 8.8%. Moreover, the data analysis of the resonance peak height of the vibrating structure made it possible to assess the changes in the deflection sensitivity. The increase in the deflection sensitivity after step 3 of the process stems from the effect of the stress concentration around the piezoresistors, which in this case received bigger stress as compared with the non-modified structure.

The deflection sensitivity of the finally modified active piezoresistive cantilever was calibrated when the thermomechanical noise vibrations were recorded and analyzed. Taking account of the resonance curves acquired using interferometers and measurement electronics it was possible to calibrate the deflection sensitivity per Volt of bridge bias voltage of 1.7 µV/nm/V and the probe stiffness of 80 N/m (Figure 5). The theoretical sensitivity of the piezoresistive beam, calculated in accordance with the procedure described in [15], was 0.8 µV/nm/V. The results of the calibration and simplified analytical calculations (sensitivity increase of 2.1) are in agreement with the results obtained experimentally (sensitivity increase of 2.5). The difference results from the assumption that in the theoretical model the cantilever beam is uniform from the material point of view. Moreover, the theoretical foundation is that the piezoresistors are placed pointwise in the area of the highest mechanical stress, whereas in real construction they are distributed along the beam. The theoretical calculations were performed on the assumption that the cantilever deflected as described in the pure bending model, whereas after the FIB modification the structure bent mostly around the piezoresistors.

## 6. Conclusions

A combination of FEM methods and experimental FIB technology were utilized to optimize the design of an active piezoresistive AFM cantilever resulting in significant increase of its deflection sensitivity. The FIB processing leads simultaneously to a slight decrease in the stiffness and the resonance frequency of the piezoresistive beam. The FIB modification technology proved to be very useful as it enabled planned modification of the piezoresistive cantilever at precisely defined locations without influencing the piezoresitors and microheater function. The applied technology of the assessment of the cantilever sensitivity based on the application of the integration with the beam deflection actuator, proved to be a useful and reliable methodology which could be used during the multistep FIB modification. After the FIB modification, the mechanical stress was concentrated around the piezoresistors, leading to the 2.5 times increase in piezoresistive bridge output (in comparison with the cantilevers without modification of the same design and parameters). These results are in agreement with the results of the beam calibration and the simplified analytical calculations. It was found that an optimal process results in improvements of the piezoresistive signal and force sensitivity, respectively. The proposed technology will enable future work including additional cantilever design optimization and new read-out schemes, as well as shrinking the dimensions to achieve higher scanning speed, consuming less power and providing higher signal to noise ratio.

## Figures and Tables

**Figure 1 sensors-19-04429-f001:**
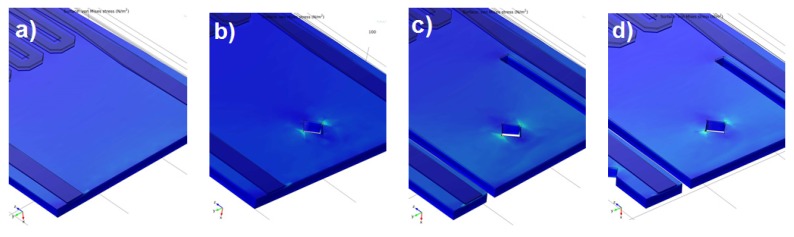
Obtained stress concentration in the area of the piezoresistive readout for the four focused ion beam (FIB) based geometry modifications: (**a**) Step 0; (**b**) step 1; (**c**) step 2; (**d**) step 3.

**Figure 2 sensors-19-04429-f002:**
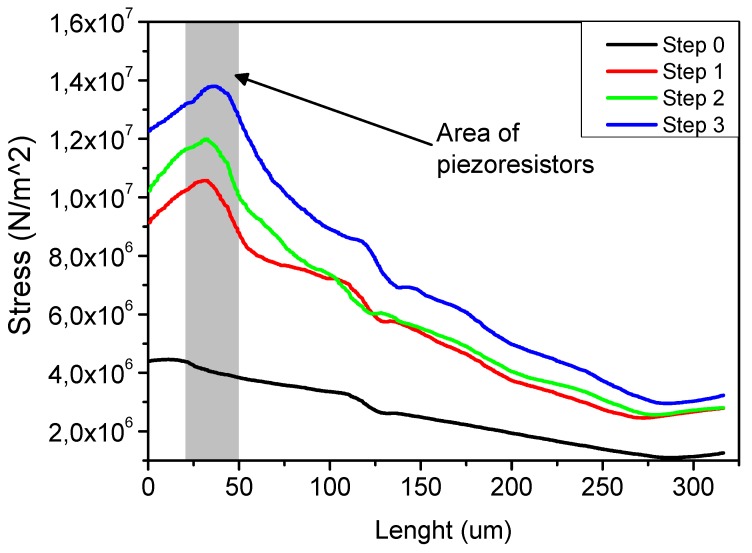
Results of the stress simulation for the active piezoresistive cantilever before and after three steps of FIB structure modification.

**Figure 3 sensors-19-04429-f003:**
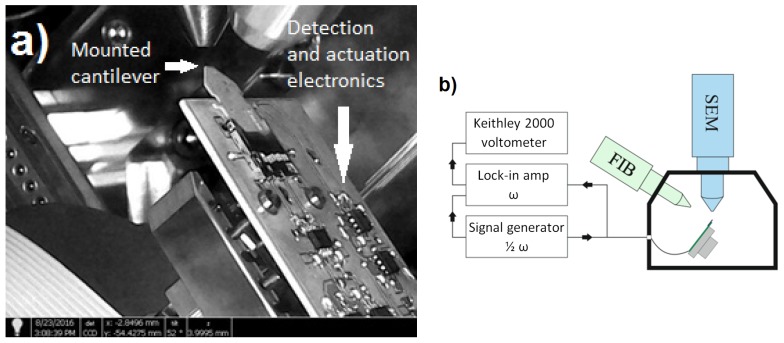
Active piezoresistive atomic force microscopy (AFM) cantilever in the FIB/SEM chamber: (**a**) Experimental setup with external signal acquisition components (**b**).

**Figure 4 sensors-19-04429-f004:**
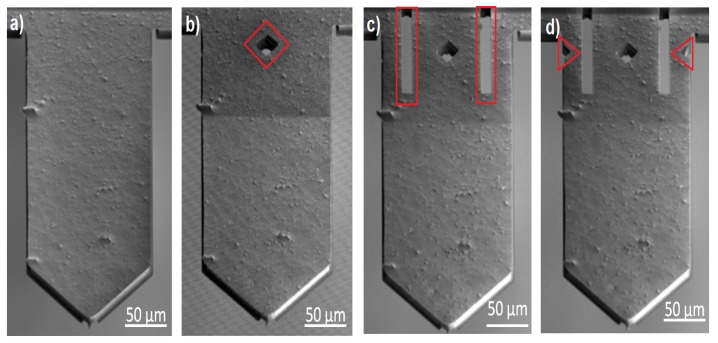
SEM image of FIB modification of the active piezoresistive cantilevers: (**a**) Step 0; (**b**) step 1; (**c**) step 2; (**d**) step 3. View from the bottom side.

**Figure 5 sensors-19-04429-f005:**
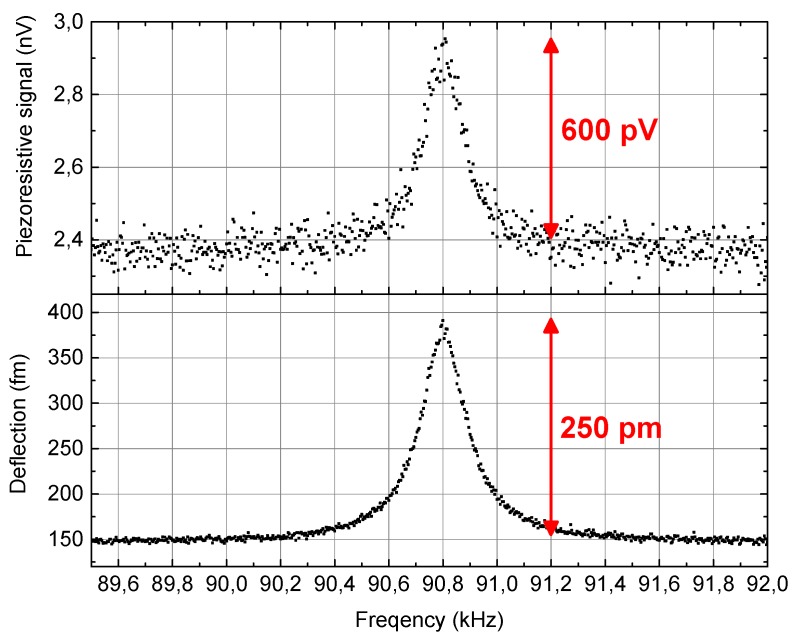
Calibration of the piezoresistive deflection detector-resonance curves of thermomechanical cantilever noise recorded using interferometer and by acquisition of the signals of the piezoresistive bridge.

**Table 1 sensors-19-04429-t001:** Material properties of the piezoresistive cantilevers.

Parameter	Al	SiO_2_	Si_3_N_4_	Si
Elastic modulus, E (MPa)	70 × 10^3^	70 × 10^3^	250 × 10^3^	170 × 10^3^
Poisson’s ratio, ν	0.35	0.17	0.23	0.28
Mass density, ρ (kg/m^3^)	2700	2200	3100	2329
Electrical conductivity, ∑ (S/m)	35.5 × 10^6^	---	1 × 10^−15^	1 × 10^−12^
Thermal conductivity, k (W/mK)	237	1.4	20	130
Coefficient of thermal expansion, α (1/K)	23.1 × 10^−6^	0.5 × 10^−6^	2.3 × 10^−6^	2.6 × 10^−6^

**Table 2 sensors-19-04429-t002:** Geometric properties of the piezoresistive cantilever.

Parameter	Dimension
Length of cantilever, L	350 μm
Width of cantilever, B	140 μm
Thickness of thermal actuator layer, t_heater_	0.7 μm
Thickness of insulation layer, t_ins_	0.5 μm

**Table 3 sensors-19-04429-t003:** FIB experiment data.

Parameter	Step 0	Step 1	Step 2	Step 3
Resonance frequency (kHz)	95.279	94.393	91.673	91.033
Resonance frequency shift (Hz)	0	886	2720	640
Stiffness (N/m)	119.94	117.71	111.03	109.49
Relative piezo signal (V\V)	1	1.3	2.05	2.54
Stress concentration simulation (N/m^2^/N/m^2^)	1	2.64	2.82	3.32

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
