# Peer review of "Sensitivity Improvement to Active Piezoresistive AFM Probes Using Focused Ion Beam Processing"

_sensors, 2019, doi:10.3390/s19204429_

Round 1

Reviewer 1 Report

This paper describes a technique for increasing the deflection sensitivity of an atomic force microscope (AFM) cantilever having an integrated piezoresistive sensor and a microheater actuator via focused ion beam (FIB) milling a hole near the base of the cantilever. The authors performed finite element modeling (FEM) of the effect of what looks like a diamond shaped hole on the stress induced on the cantilever and observed a 2.5 X improvement in the deflection sensitivity of the cantilever while suffering only minimal softening of the cantilever. The authors then fabricated the cantilever probe with a similar hole geometry and made observations on the cantilever performance that agreed with their modeling predictions.

Ever since the invention of the AFM, researchers have been integrating a variety of sensors to improve the functionality of AFM probes, obviate the need for the traditional optical beam-based setup found in commercial AFMs and scale to arrays of cantilevers for parallel sensing and improved throughput. Furthermore, several researchers have implemented a variety of so-called self-actuation techniques and actuators that provide precise and targeted actuation of cantilevers. The research from C. F. Quate, P. Vettiger, W. P. King, C. Liu, S. C. Minne, P. K. Hansma, M. Favre, etc. come to mind. In light of past research, I find the research presented in this paper to be modest in its significance and incremental in nature. As it stands, I find this paper lacking in innovativeness, scientific rigor, clarity, professionalism, and polish to earn publication in Sensors. Here are my chief concerns:

Since this paper is focused around the geometry of the cantilever, the authors need to explain the structure and geometry in greater detail, preferably with figures, even if the details are referenced. The authors need to describe the actuator in greater detail. A reader not familiar with their previous work is unlikely to understand their actuator. On some occasions it is referred to as a “microheater” and on other occasions as a plain actuator. The authors need to explain their optimization process for arriving at their so-called optimal geometry for the cantilever. The paper suspiciously reads as though the authors first experimentally fabricated the modified cantilever with a pre-determined geometry and then developed an FEM that agreed with their experimental results. I have personally observed several researchers doing this. The authors do not even describe the so-called optimal geometry. There are no figures that clearly illustrate the dimensions of the cantilever and the modifications. The authors need to reconcile the broader impacts of this work in a discussion section. For example, integrating FIB milling into an otherwise typical CMOS fabrication process prevents the fabrication from being scalable. Could the authors not have changed the photolithography masks to introduce the hole? Was this project focused on creating one-off cantilevers from the very beginning? What impact does any ion implantation from the FIB milling have on the cantilever’s performance? What needs to be done for such cantilevers to gain main-stream acceptance in the industry? Etc. The authors need to explain why they have not calibrated the cantilevers in an AFM. IF they have, do those results match with what are presented in the table(s)? Can the authors demonstrate the benefits of the improved sensitivity by using the modified cantilevers in some real or calibration experiments in an AFM? The authors describe four steps in the FIB milling process. However, the figures and the accompanying text do not clearly explain the differences in these so-called steps. I cannot make sense of the differences in the “steps” presented in figures 2 and 4. Are the author sure of the stiffness numbers quoted in table 3? They seem a few orders of magnitude higher than they typically are for cantilevers their size. Can the authors explain the wide variations in the thickness of the cantilevers (2.5 um to 10 um)? I request the authors to stay consistent with the formatting of their numbers. On certain occasions, the authors say that the used “11,000 elements” in their mesh and at the same time, they use “0,9 um” for dimensions. Please use the “,” and “.” In a consistent manner. This paper lacks the basic professionalism and polish necessary for a published academic paper. First, I find that the authors did not even proof-read their paper – please see lines 67 – 78 on page 2. Second, there are several grammatical and spelling errors. I request that the authors have the paper proof-read by a native English speaker. Third, certain equations are numbered while others are not. Fourth, the authors describe the fabrication of the basic cantilever and leave out the FIB milling in section 3. Given that the authors present the modeling results before the fabrication, the sections should flow consistently. Fifth, some variables in equation (2) remain unexplained. Sixth, figures 1, 3, and 4 need more labels for readers to make sense of them.

Reviewer 2 Report

This work presents the sensitivity improvement by modifying the AFM beam profile using FIB. However, the interpretation of increase principal is not clear. The writing and organization of the manuscript are very poor. 

1) Section 1, the final paragraph is not related with the research, which should be deleted.

2) Section 2 is absence.

3) Section 5 and 6 have the same titles.

4) Many spelling mistakes.

Round 2

Reviewer 2 Report

It seems the authors made efforts on improving the writing of the manuscript. I suggest the authors may change the procedure of sections. Section 3 (Theoretical background) and 4 (modelling and simulation)  can be changed to section 2 and 3. The style of Table A1 should be improved. The unit of vertical axis  in Figure A3 should be corrected as nV or fm.

Dot should be used in the expression of numeral instead of comma in the full manuscript. 

Author Response

Dear Reviewer 2,

I would like to kindly thank your for all given remarks. In the uploaded new version of the manuscript you may find the corrections. I hope they made the paper more transparent for readers.

Best regards,

Piotr Kunicki